# MEMORY-DRIVEN MULTIMODAL CHAIN OF THOUGHT FOR EMBODIED LONG-HORIZON TASK PLANNING

## ABSTRACT

Existing methods excel in short-horizon tasks but struggle with complex, long-horizon planning in dynamic environments. To address these limitations, we propose the Memory-Driven Multimodal Chain of Thought (MCoT-Memory), a framework designed to enhance task planning through two key innovations: 1) Evolving Scene Graph-Driven Chain of Thought with CoT Memory Retrieval, which enables the agent to continuously update a scene graph with visual information captured along its trajectory, providing a structured and dynamic representation of the environment that informs real-time decision-making, and uniquely incorporates CoT memory retrieval to allow the agent to leverage past experiences in its reasoning process; 2) Stepwise Confidence-Driven Memory Retention, which employs an expert model to evaluate reasoning across multiple dimensions of accuracy, ensuring that only high-confidence experiences are retained in memory for future retrieval, thus enabling the agent to build on valuable insights and improve performance in long-horizon tasks. To advance long-horizon task planning, we present ExtendaBench, a comprehensive benchmark encompassing 1,198 tasks across two simulators, VirtualHome and Habitat 2.0. The tasks are categorized into ultra-short, short, median, and long tasks. Extensive experiments demonstrate that prior methods struggle with long-horizon tasks, while MCoT-Memory significantly improves performance, marking it as a promising approach for embodied task planning.

## 1 INTRODUCTION

In the domain of autonomous systems, the expectation for robots to execute complex, real-world tasks in domestic settings has significantly increased. Tasks such as organizing a room, preparing a meal, and cleaning up afterward require not only diverse actions but also long-term planning. However, current approaches struggle with long-horizon tasks due to limited research in this area and the dominance of benchmarks Puig et al. (2018); Liao et al. (2019); Shridhar et al. (2020a;b) focused on short, discrete tasks. This gap hinders progress toward robots capable of handling the complex, multi-step tasks demanded by real-life scenarios.

The advent of Large Language Models (LLMs) OpenAI (2023); Touvron et al. (2023); Chiang et al. (2023); Geng et al. (2023) has led to notable advancements in task planning. Several approaches have leveraged LLMs to determine subsequent actions within task sequences. Some methods Huang et al. (2022a); Ahn et al. (2022) score potential actions based on their alignment with LLM-predicted outcomes, while others Huang et al. (2022b) use LLMs to directly generate actions. Additionally, studies Huang et al. (2022b); Singh et al. (2023); Wake et al. (2023); Bhat et al. (2024) have integrated environmental feedback to enhance adaptability in dynamic conditions. However, approaches like ReAct Yao et al. (2022), which employ Chain-of-Thought (CoT) reasoning, are limited by their single-modality (text) input and lack of a memory mechanism. On the other hand, methods like RAP Kagaya et al. (2024) focus more on memory but still depend heavily on external rewards to store successful experiences, which restricts their ability to autonomously explore and learn. Both methods are less suited to long-horizon tasks that require the integration of multimodal information and autonomous self-improvement, which are essential for robots in complex environments.

To address the limitations of existing methods in long-horizon task planning, we propose Memory-Driven Multimodal Chain of Thought (MCoT-Memory), a framework designed to tackle the chal-

lenges of complex task planning in dynamic environments. Our approach introduces two key innovations: (1) Evolving Scene Graph-Driven CoT: This component allows the agent to continuously update a scene graph with visual information captured along its trajectory. The evolving scene graph provides a structured and dynamic representation of the environment, enabling the agent to make decisions based on real-time context. Unlike prior methods that rely on static or text-based inputs, our approach leverages the visual dynamics of the agent's surroundings to inform its reasoning. (2) Stepwise Confidence-Driven Memory Retention: After task completion, an expert model evaluates each reasoning step based on coherence, relevance, common-sense alignment, and overall task completion. The aggregated score determines whether the entire reasoning process is stored in memory. This ensures that only high-confidence reasoning processes are retained, allowing the agent to reuse valuable experiences in future tasks and improving its performance on long-horizon tasks. By integrating these two innovations, MCoT-Memory enables more effective long-horizon task planning, combining dynamic visual updates with selective memory retention to address the challenges of real-world, multi-step tasks.

Finally, addressing the notable gap in the field regarding the absence of a benchmark tailored for long-horizon tasks, we propose a comprehensive benchmark ExtendaBench divided into four categories based on the number of steps required to complete the tasks: ultra-short, short, median, and long. Utilizing the generative capabilities of GPT-4 OpenAI (2023), we produced a vast array of tasks. These tasks underwent minimal human correction to ensure high-quality data while substantially reducing the cost associated with manual data labeling. To validate the efficacy of our approach, we conducted comparative analyses against several baselines within this newly proposed benchmark. The results unequivocally demonstrate that our method significantly enhances accuracy.

The contributions of this work are summarized as follows:

- We introduce MCoT-Memory, a novel framework that combines evolving scene graph-driven reasoning with stepwise confidence-driven memory retention, enabling robots to handle long-horizon, multi-step tasks in dynamic environments more effectively.
- We propose a challenging benchmark, ExtendaBench, comprising four distinct sets that collectively include 1,198 tasks. This benchmark is designed for evaluating long-horizon tasks, providing a comprehensive platform for testing task-planning models.
- We implement several baselines and validate the effectiveness of MCoT-Memory. Extensive experimental results showcase the considerable enhancements attributed to MCoT-Memory.

## 2 RELATED WORK

### 2.1 MULTIMODAL LARGE LANGUAGE MODELS

The emergence of LLMs Touvron et al. (2023); Chiang et al. (2023) has driven substantial progress in multimodal large language models (MLLMs), which aim to integrate both visual and textual modalities, advancing toward a more generalized form of intelligence. Early works such as BLIP-2 Jian et al. (2024), MiniGPT-4 Zhu et al. (2023), LLaVA Liu et al. (2024b), and OpenFlamingo Awadalla et al. (2023) capitalized on pretrained vision encoders paired with LLMs, demonstrating strong performance in tasks like visual question answering and image captioning. mPLUG-Owl Ye et al. (2023) introduces a modularized training framework to further refine cross-modal interactions. On the closed-source side, models such as GPT-4V OpenAI (2023) and Gemini Team et al. (2023) exemplify some of the most advanced MLLMs, pushing the boundaries of multimodal reasoning and interaction capabilities.

### 2.2 CHAIN OF THOUGHT

Recent advancements in natural language processing have highlighted the effectiveness of LLMs in employing Chain-of-Thought (CoT) reasoning to improve complex problem-solving. CoT techniques encourage models to explicitly outline intermediate steps in reasoning, which has been shown to significantly enhance their cognitive abilities Wei et al. (2022); Kojima et al. (2022). Ongoing research efforts have explored various approaches, such as optimizing the selection of examples Rubin et al. (2021); Lu et al. (2022); Fu et al. (2022), integrating programming tasks Chen et al. (2022),

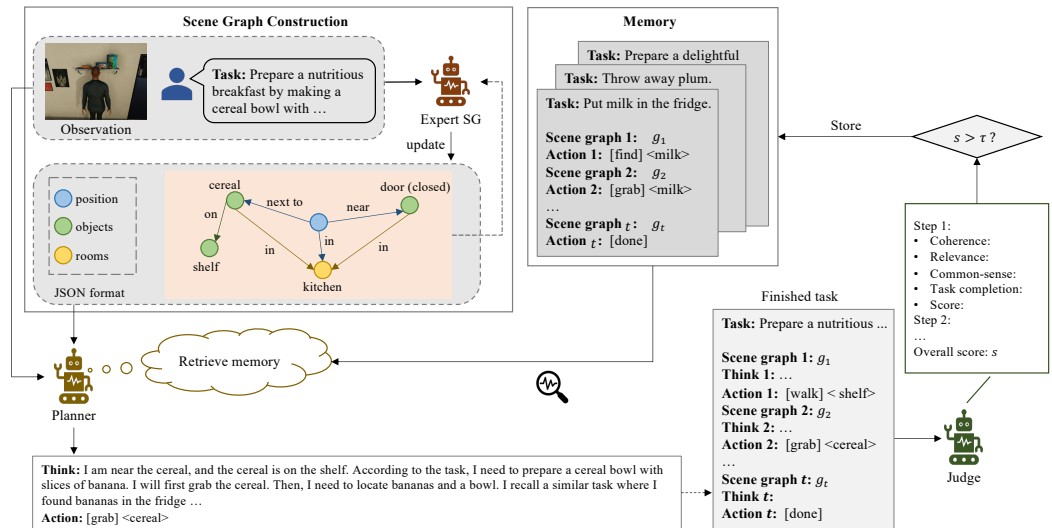

Figure 1: The overview of our proposed MCoT-Memory, where the expert model (Expert SG) generates the scene graph based on the task and observations. The planner then performs Chain-of-Thought (CoT) reasoning using the task, constructed scene graph, and similar past experiences. A judge evaluates each reasoning step and determines whether to store the process in the memory pool for future retrieval.

decomposing problems into smaller steps Khot et al. (2022); Zhou et al. (2022), and calibrating rationales for improved consistency Wang et al. (2022); Li et al. (2022b). In the realm of multi-modal research, Zhang et al. (2023) incorporates visual information to enhance reasoning capabilities. Other methods build on this by introducing sub-question decomposition Zheng et al. (2023); Jiang et al. (2024), contrastive comparison techniques Zhang et al. (2024), scene graph generation for structured visual understanding Mitra et al. (2024), and the use of multimodal hybrid rationales for more comprehensive reasoning Zhou et al. (2024).

## 2.3 EMBODIED TASK PLANNING

Traditional robotics planning methods have relied on search algorithms in predefined domains Fikes & Nilsson (1971); Garrett et al. (2020); Jiang et al. (2018), but face scalability challenges in complex environments with high branching factors Puig et al. (2018); Shridhar et al. (2020a). Heuristics have helped alleviate these limitations, leading to advancements Baier et al. (2009); Hoffmann (2001); Helmert (2006); Bryce & Kambhampati (2007). More recently, learning-based methods like representation learning and hierarchical strategies have emerged, showing effectiveness in complex decision-making Eysenbach et al. (2019); Xu et al. (2018; 2019); Srinivas et al. (2018); Kurutach et al. (2018); Nair & Finn (2019); Jiang et al. (2019).

The advent of LLMs has further revolutionized planning by enabling task decomposition and robust reasoning Li et al. (2022a); Huang et al. (2022a;b); Ahn et al. (2022); Valmeekam et al. (2022); Silver et al. (2022); Song et al. (2023); Rana et al. (2023); Driess et al. (2023); Liu et al. (2023b); Wu et al. (2023); Wake et al. (2023); Chen et al. (2023); Qiu et al. (2023); Bhat et al. (2024); Zhi-Xuan et al. (2024). Other works focus on translating natural language into executable code and formal specifications Vemprala et al. (2023); Singh et al. (2023); Liang et al. (2023); Silver et al. (2023); Xie et al. (2023); Skreta et al. (2023); Liu et al. (2023a); Zhang & Soh (2023); Ding et al. (2023b;a); Zhao et al. (2024). Some approaches fine-tune LLMs for better performance Ahn et al. (2022); Driess et al. (2023); Qiu et al. (2023), while others opt for few-shot or zero-shot methods to avoid the resource demands of model training.

## 3 MCoT-Memory

We present the Memory-Driven Multimodal Chain of Thought (MCoT-Memory) framework, designed for long-horizon task planning in dynamic environments. Our approach introduces two key innovations: the Evolving Scene Graph-Driven CoT, which enables real-time updates of a task-related scene graph that focuses on key objects, and the Stepwise Confidence-Driven Memory Retention, which selectively stores high-confidence reasoning processes for future tasks. The following sections will detail each component and its implementation within the framework.

### 3.1 Evolving Scene Graph-Driven CoT

This module initiates with the construction of the scene graph, establishing a structured representation of the environment. Subsequently, the agent generates actions based on this scene graph, relevant observations, and associated memory through CoT reasoning. Finally, the scene graph undergoes dynamic updates to reflect environmental changes, ensuring the agent's understanding remains accurate and current.

#### 3.1.1 Initial Scene Graph Construction

The initial construction of the scene graph is pivotal for structuring and preserving visual information captured during the robot's task execution. By systematically representing the environment, the agent is enabled to effectively reason about its surroundings and make informed decisions. We employ a MLLM-based expert, such as LLAVA, to generate the scene graph $g_1$ based on the task description $T$, visual observation $o_1$, and a prompt specifically tailored for scene graph generation $P_{\text{SG}}$ in the first step:

$$g_1 = \mathcal{E}_{\text{SG}}(o_1, T, P_{\text{SG}}). \tag{1}$$

The scene graph $g_1$ is structured in a format like JSON, and consists of five key attributes:

- *Position: The robot's location relative to key objects or rooms.*
- *Objects: Key entities present in the scene, with their positions and states.*
- *Rooms: Spaces observed by the robot during task execution.*
- *States: Conditions of objects in the environment.*
- *Relationships: Spatial and relational connections between objects and rooms.*

#### 3.1.2 Action Generation through CoT Reasoning

The MLLM subsequently generates actions grounded in rationales derived from observations, the scene graph, task descriptions, and specific reasoning prompts. This two-step process enhances the quality of action generation by ensuring each action is underpinned by a clear rationale. In each step $i$, the agent produces rationales $r_i$ by evaluating the inputs: observation $o_i$, the scene graph $g_i$, task description $T$, and the CoT reasoning prompt $P_{\text{CoT}}$. A typical prompt for CoT reasoning might be:

*Based on the provided information, think step-by-step to identify key factors for deciding the next action.*

This can be expressed as:

$$r_i, a_i = f(o_i, g_i, T, P_{\text{CoT}}), \tag{2}$$

where $r_i$ encapsulates the reasoning behind potential actions, linking the current environmental state to task objectives, and $a_i$ is the predicted next action.

Additionally, relevant memories $M'$ (discussed in later sections) can inform both rationale and action generation, enriching the decision-making process. Insights from prior experiences enhance the model's ability to approach similar tasks effectively:

$$r_i, a_i = f(o_i, g_i, T, M', P_{\text{CoT}}). \tag{3}$$

#### 3.1.3 Dynamic Scene Graph Updates

As the agent navigates its environment and executes tasks, the scene graph must be continuously updated to reflect changes in surroundings and task context. This dynamic updating process is

critical for maintaining an accurate environmental representation, directly influencing the agent's reasoning and decision-making capabilities. To facilitate these updates, we implement a feedback loop that integrates new observation $o_{i+1}$ with the existing scene graph $g_i$. The updated scene graph $g_{i+1}$ can be expressed as:

$$g_{i+1} = \mathcal{E}_{\text{SG}}(o_{i+1}, T, g_i, P_{\text{SG}}), \tag{4}$$

By updating all key attributes dynamically, the scene graph remains a reliable and current representation of the environment, ensuring that the agent's decision-making process is based on accurate and up-to-date information.

### 3.2 STEPWISE CONFIDENCE-DRIVEN MEMORY BANK

The Stepwise Confidence-Driven Memory Bank stores high-confidence reasoning processes from completed tasks. It selectively retains valuable experiences, including task descriptions, scene graphs, reasoning steps, and actions. This memory is then used to guide decision-making in future tasks. The following sections cover how experiences are stored and how relevant experiences are retrieved.

#### 3.2.1 EVALUATING COT PROCESSES FOR MEMORY RETENTION

In the Stepwise Confidence-Driven Memory Bank, after completing a task $T$, the entire CoT process $\{(r_1, a_1), (r_2, a_2), \ldots, (r_t, a_t)\}$ is evaluated by a MLLM-based judge model $\mathcal{E}_{\text{eval}}$. This expert model evaluates the task based on criteria such as coherence, relevance, common-sense alignment, and task completion, ensuring a thorough assessment of the CoT process. The expert model takes as input the full task description, the reasoning steps, and a specific evaluation prompt $P_{\text{eval}}$, which defines the following criteria:

- *Coherence: Ensuring logical consistency throughout the CoT steps.*
- *Relevance: Verifying that each step is directly applicable to the current task.*
- *Common-Sense Alignment: Assessing whether the steps adhere to basic real-world knowledge.*
- *Task Completion: Evaluating the effectiveness of the reasoning in achieving the task's goal.*

Based on these criteria, the expert model evaluates the entire CoT process and outputs the score and justification for each reasoning step, along with a final overall score for the task, which reflects:

1. *The cumulative effectiveness of all reasoning steps combined.*
2. *The overall task completion, including whether the robot achieved the intended goal.*

This evaluation can be expressed as:

$$(j_1, s_1), (j_2, s_2), \ldots, (j_t, s_t), s = \mathcal{E}_{\text{eval}}(T, \{(r_1, a_1), (r_2, a_2), \ldots, (r_t, a_t)\}, P_{\text{eval}}), \tag{5}$$

where $j_i$ is the justification for the score, $s_i$ is the score for step $r_i$, and $s$ is the final score for the entire task. If $s$ exceeds a predefined threshold $\tau$, the task-specific elements are added to the memory pool $M$ as follows:

$$M \leftarrow M \cup \{(T, \{(g_1, r_1, a_1), \ldots, (g_t, r_t, a_t)\})\}. \tag{6}$$

#### 3.2.2 RETRIEVING SIMILAR EXPERIENCES FROM MEMORY BANK

When retrieving similar experiences from the memory pool $M$, where $M$ has a length of $L$, the objective is to compute the similarity between the current task $T'$ and the stored experiences in $M$. Additionally, for each step $i$ in the current task, the scene graph $g'_i$ is compared with the final scene graph $g_{l,t}$ from each stored experience. We utilize sentence-transformers Reimers & Gurevych (2019) to compute the similarity for both the task descriptions and the scene graphs. The formula for computing the total similarity between the current task $T'$ and a stored experience $(T_l, g_{l,t})$ (where $l = 1, \ldots, L$) at step $i$ is given by:

$$\mathcal{S}(T', g'_i, T_l, g_{l,t}) = \lambda_1 \cdot \text{sim}(T', T_l) + \lambda_2 \cdot \text{sim}(g'_i, g_{l,t}), \tag{7}$$

where $\lambda_1$ and $\lambda_2$ are the weighting factors that control the relative importance of task similarity and scene graph similarity. After calculating the scores for all stored experiences, the top $k$ experiences with the highest similarity scores are retrieved:

$$\{(T_l, g_l, r_l, a_l)\}_{l \in \arg \text{top}_k \mathcal{S}(T', g'_i, T_l, g_{l,t})}. \tag{8}$$

By using this method, the agent retrieves relevant experiences from the memory pool, taking into account both the task description and the scene graph at each step of the current task.

# 4 EXTENDABENCH

The ExtendaBench task corpus was developed using two distinct approaches tailored to each simulator. For VirtualHome Puig et al. (2018), we leveraged GPT-4's advanced generative capabilities to create diverse and complex tasks. In contrast, tasks for Habitat 2.0 Szot et al. (2021) were systematically collected using pre-defined templates.

## 4.1 VIRTUALHOME

The creation of the ExtendaBench task corpus for VirtualHome harnesses GPT-4's powerful generative abilities to produce diverse tasks. The process of generating tasks can be divided into three stages: generation, review, and refinement.

### 4.1.1 GENERATION

The initial phase begins within the confines of VirtualHome, a simulated environment, where a varied collection of objects sets the stage for a multitude of task scenarios. By employing GPT-4 as the task generator, we design tasks focused on object manipulation, striving for a wide array of task varieties and complexities. This method ensures an exhaustive representation of scenarios that closely mimic real-world challenges. To facilitate the generator's task creation, we provide prompts that are carefully constructed to inspire a broad range of tasks. An illustrative example of such a prompt is as follows:

*Given an "HUMAN ACTION LIST" and an "OBJECT LIST", you need to use some of them to compose a new household task. And then generate a description of the task followed by decomposing the task into steps.*

### 4.1.2 REVIEW

In the subsequent phase, GPT-4 undertakes the generation of detailed action plans for the devised tasks, meticulously outlining the steps required for successful task execution. To ensure the feasibility and coherence of these tasks, we introduce an additional examiner of scrutiny, also powered by GPT-4. This examiner evaluates each task and its associated action plan for clarity, necessity, and coherence of steps, as well as the relevance and practicality of the actions and items involved, ensuring they belong to the simulated environment VirtualHome. It also assesses each step for common sense applicability, providing constructive feedback for further refinement. Below is an illustrative prompt that could be used to guide the examiner in its evaluation role:

*Given a task with its decomposed steps, an "HUMAN ACTION LIST" and an "OBJECT LIST", you need to check that the actions and objects in the decomposed steps are all in the given "HUMAN ACTION LIST" and "OBJECT LIST".*

### 4.1.3 REFINEMENT

After undergoing expert scrutiny, the generator refines the tasks and their corresponding action plans. Subsequent simulation of these revised tasks and plans enables further improvements based on simulator feedback. Tasks that are successfully executed within the simulator receive preliminary approval. Nevertheless, to guarantee optimal quality and applicability, we subject each task to a rigorous manual review, evaluating them for practicality and realism. Tasks that do not achieve success in the simulation are minimally modified bu human according to the simulator's feedback, focusing on enhancing their realism and feasibility. Below is the prompt for refining according to feedback from simulator are as below:

*Analyze the reasons for the failed steps and determine if the task is feasible under the given rules. If feasible, output your reasoning and suggest modifications to the failed steps.*

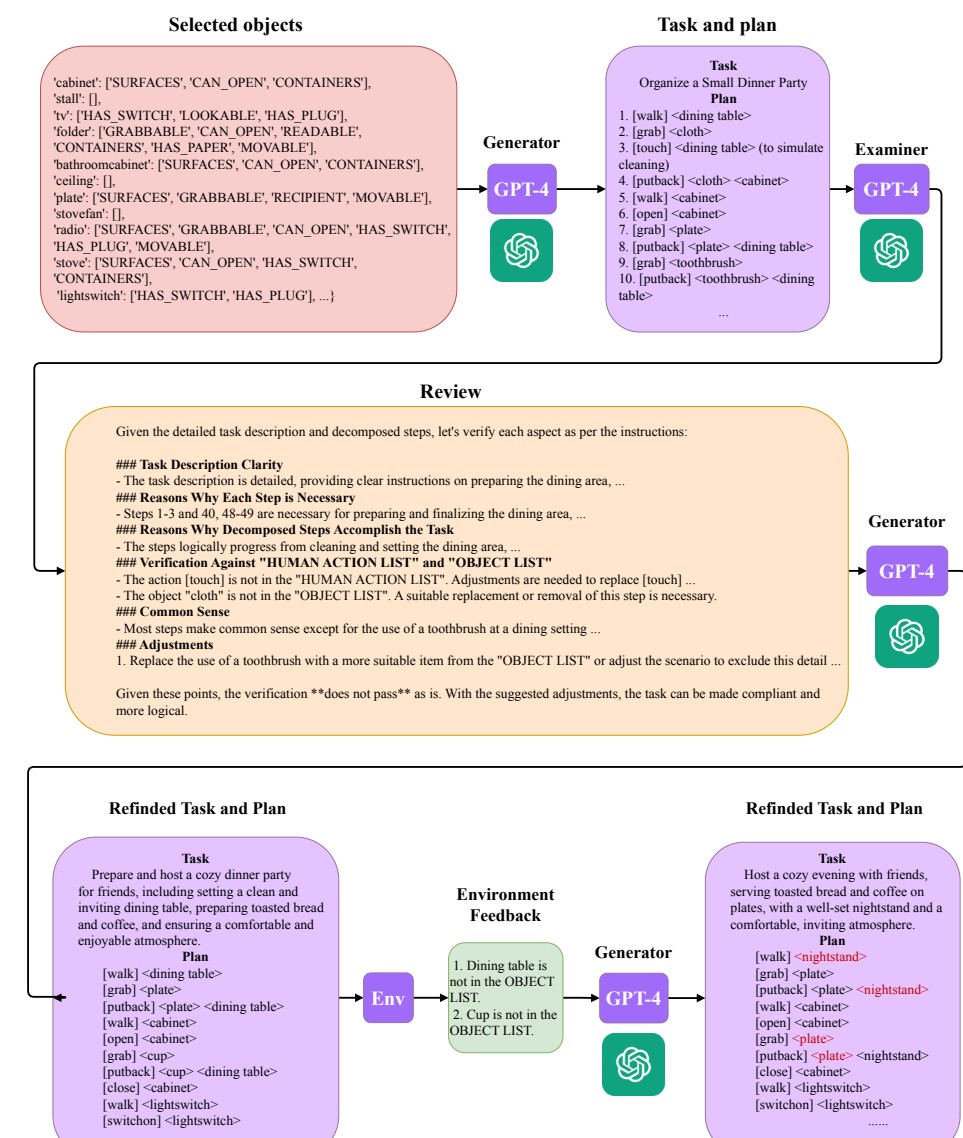

Figure 2: The process of generating tasks in ExtendaBench.

The multi-stage process, with minimal human intervention, is designed to ensure the reliability and quality of the tasks and their associated plans. This methodology reduces inaccuracies and ensures that ExtendaBench represents a broad range of complex, real-world tasks. The whole process of generating tasks in benchmark is shown in Figure 2.

## 4.2 HABITAT 2.0

Following the Language Rearrangement Szot et al. (2023), we utilized predefined templates to generate tasks for Habitat 2.0. However, in contrast to their method, we significantly extended the length of the action sequences, enabling the evaluation of long-horizon planning algorithms on more complex and extended tasks. This modification allows for a more thorough assessment of an agent's ability to handle diverse and challenging environments.

Table 1: Compare with existing methods on different sets of our ExtendaBench on VirtualHome.

| Method | Ultra-Short | | Short | | Median | | Long | | Average | |
|---|---|---|---|---|---|---|---|---|---|---|
| | GCR | SR | GCR | SR | GCR | SR | GCR | SR | GCR | SR |
| InternVL2-26B | 55.76 | 33.33 | 24.92 | 0.00 | 19.87 | 0.00 | 24.18 | 0.00 | 31.18 | 8.33 |
| LLaVa-v1.6-34B | 61.39 | 30.00 | 30.18 | 0.00 | 18.38 | 0.00 | 22.01 | 0.00 | 32.99 | 7.50 |
| LLaVa-v1.6-34B (CoT) | 61.72 | 30.00 | 34.88 | 0.00 | 21.61 | 0.00 | 22.34 | 0.00 | 35.14 | 7.50 |
| LLaVa-v1.6-34B (CCoT) | 64.25 | 30.00 | 27.25 | 0.00 | 23.43 | 0.00 | 22.33 | 0.00 | 34.31 | 7.50 |
| LLaVa-v1.6-34B (DDCoT) | 63.48 | 26.67 | 29.33 | 0.00 | 23.87 | 0.00 | 22.86 | 0.00 | 34.89 | 6.67 |
| LLaVa-v1.6-34B (MCoT-Memory) | 69.31 | 43.33 | 42.48 | 3.33 | 25.84 | 0.00 | 29.11 | 0.00 | 41.68 | 11.67 |
| GPT-4o | 82.31 | 46.67 | 82.22 | 26.67 | 63.17 | 13.33 | 50.40 | 6.67 | 69.52 | 23.33 |
| GPT-4o (MCoT-Memory) | 85.09 | 50.00 | 83.58 | 36.67 | 74.70 | 26.67 | 60.80 | 6.67 | 76.04 | 30.00 |

Table 2: Compare with GPT-4o on different sets of our ExtendaBench on Habitat.

| Method | Ultra-Short | | Short | | Median | | Long | | Average | |
|---|---|---|---|---|---|---|---|---|---|---|
| | GCR | SR | GCR | SR | GCR | SR | GCR | SR | GCR | SR |
| GPT-4o | 51.11 | 53.33 | 22.65 | 33.33 | 18.48 | 0.00 | 9.78 | 0.00 | 25.51 | 21.67 |
| GPT-4o (MCoT-Memory) | 53.54 | 55.00 | 25.01 | 26.67 | 19.04 | 0.00 | 11.14 | 0.00 | 27.18 | 20.42 |

## 4.3 DATASET STATISTICS

The categorization within ExtendaBench is defined by the length of the action sequence required to accomplish a task, distributed as follows:

- Ultra-Short Tasks: Tasks that can be completed in fewer than 10 actions.
- Short Tasks: Tasks requiring 10 to 20 actions for completion.
- Medium Tasks: Tasks necessitating 20 to 30 actions to finish.
- Long Tasks: Tasks that demand more than 30 actions to complete.

The VirtualHome set includes a total of 294 tasks, with 103 ultra-short tasks, 65 short tasks, 69 medium tasks, and 57 long tasks. Similarly, the Habitat 2.0 set comprises 904 tasks, distributed as 161 ultra-short tasks, 243 short tasks, 190 medium tasks, and 310 long tasks.

## 5 EXPERIMENTS

### 5.1 EXPERIMENTAL SETUP

For the VirtualHome set, we used 120 tasks as the test set (30 tasks from each category), with the remaining tasks serving as the training set, which can also be used as prompts. Similarly, the Habitat 2.0 set also includes 120 test tasks. To assess system efficacy, we employ success rate (SR) and goal conditions recall (GCR) as our primary metrics. SR measures the proportion of executions where all key goal conditions, identified as those that change from start to finish during a demonstration, are met. GCR calculates the discrepancy between the expected and achieved end state conditions, relative to the total number of specific goal conditions needed for a task. A perfect SR score of 1 corresponds to achieving a GCR of 1, indicating flawless task execution. Results of SR and GCR are both reported in %.

### 5.2 COMPARE WITH PREVIOUS METHODS

#### 5.2.1 VIRTUALHOME

**Baseline Performance:** We compared the performance of two state-of-the-art MLLMs: InternVL2-26B Chen et al. (2024) and LLaVa-v1.6-34B Liu et al. (2024a), across four task sets in our ExtendaBench benchmark, as shown in Table 1. The results demonstrate that LLaVa v1.6-34B outperforms InternVL2-26B in terms of GCR on the Ultra-Short and Short task sets, with a slightly higher average GCR across all four task sets, establishing it as a stronger baseline for multimodal task reasoning.

Table 3: Ablation studies of diferent modules in our MCoT-Memory on VirtualHome. ESG indicates evolving scene graph, while Memory represents stepwise confidence-driven memory bank.

| ESG | CoT | Memory | Ultra-Short | | Short | | Median | | Long | | Average | |
|---|---|---|---|---|---|---|---|---|---|---|---|---|
| | | | GCR | SR | GCR | SR | GCR | SR | GCR | SR | GCR | SR |
| ✗ | ✗ | ✗ | 61.39 | 30.00 | 30.18 | 0.00 | 18.38 | 0.00 | 22.01 | 0.00 | 32.99 | 7.50 |
| ✓ | ✗ | ✗ | 58.66 | 33.33 | 42.65 | 0.00 | 22.89 | 0.00 | 26.44 | 0.00 | 37.66 | 8.33 |
| ✓ | ✓ | ✗ | 65.42 | 40.00 | 40.53 | 3.33 | 24.75 | 0.00 | 27.75 | 0.00 | 39.61 | 10.83 |
| ✓ | ✓ | ✓ | 69.31 | 43.33 | 42.48 | 3.33 | 25.84 | 0.00 | 29.11 | 0.00 | 41.68 | 11.67 |

**Results of CoT Variants:** Using LLaVa-v1.6-34B as the baseline, we implemented three Chain of Thought (CoT) variants: standard CoT Wei et al. (2022), Compositional CoT (CCoT Mitra et al. (2024)), and Duty-Distinct CoT (DDCoT Zheng et al. (2023)). While CCoT and DDCoT demonstrated improvements in the GCR metric, with CCoT achieving 34.31% and DDCoT reaching 34.89%, they did not surpass the performance of standard CoT in task planning scenarios. These results suggest that CCoT and DDCoT are less suitable for the task planning tasks in our benchmark.

**Comparison with CoT Variants:** Our proposed MCoT-Memory framework demonstrated significant improvements over the baseline and other CoT variants, particularly in terms of both GCR and SR, as shown in Table 1. MCoT-Memory achieved the highest GCR and SR across all task sets, with an average GCR of 41.68% and SR of 11.67%, surpassing the performance of standard CoT, CCoT, and DDCoT. These results highlight the effectiveness of MCoT-Memory in addressing long-horizon task planning by leveraging memory retrieval and evolving scene graph-driven reasoning. Its superior performance underscores its robustness in complex task planning.

**Results on GPT-4o:** In addition to our comparisons with existing methods, we evaluated the performance of GPT-4o and our enhanced version, GPT-4o (MCoT-Memory), across all task categories. As shown in Table 1, GPT-4o (MCoT-Memory) consistently outperforms the standard GPT-4o model in both GCR and SR metrics. These results highlight the effectiveness of the MCoT-Memory framework in leveraging dynamic memory retention and evolving scene graph reasoning, leading to superior task completion performance across all task horizons.

### 5.2.2 HABITAT 2.0

We also compare the performance of GPT-4o and GPT-4o (MCoT-Memory) on different task sets from the ExtendaBench benchmark using the Habitat environment. As shown in Table 2, GPT-4o (MCoT-Memory) consistently outperforms GPT-4o across various categories on GCR. Although the average success is slightly low, the improvement in GCR suggests that MCoT-Memory is better at understanding and recalling important task details even in longer, more challenging tasks.

### 5.3 ABLATION STUDY

To further investigate the contributions of different components in our MCoT-Memory framework, we conducted a series of ablation studies, as shown in Table 3. We ablated the evolving scene graph-driven CoT and the stepwise confidence-driven memory bank modules to assess their impact on the model's performance.

**Impact of Evolving Scene Graph:** The first row in Table 3 represents the baseline, where only LLaVa v1.6-34B is used without any additional modules. The second row introduces the evolving scene graph (ESG) module. With ESG providing dynamic updates during task execution, the model shows a clear improvement, achieving an average GCR of 37.66% and SR of 8.33%. The improvement demonstrates the benefit of incorporating dynamic scene information for enhanced task understanding and execution.

**ESG-Driven CoT:** The third row represents the combination of ESG and CoT reasoning, forming the ESG-driven CoT method. This setup further enhances performance, reaching an average GCR of 39.61% and SR of 10.83%. The CoT reasoning, together with the evolving scene updates, allows the model to process more complex tasks, as evidenced by the improvements in the Short and Median task sets.

Table 4: Comparison of memory retention methods on VirtualHome: evaluating each step of the CoT versus evaluating the entire plan as a whole.

| | Ultra-Short | | Short | | Median | | Long | | Average | |
|---|---|---|---|---|---|---|---|---|---|---|
| Evaluation | GCR | SR | GCR | SR | GCR | SR | GCR | SR | GCR | SR |
| entire plan | 59.42 | 13.33 | 39.98 | 3.33 | 22.49 | 0.00 | 27.26 | 0.00 | 37.29 | 4.17 |
| each step | 69.31 | 43.33 | 42.48 | 3.33 | 25.84 | 0.00 | 29.11 | 0.00 | 41.68 | 11.67 |

**Full MCoT-Memory Framework:** The fourth row in Table 3 corresponds to the complete MCoT-Memory framework, which integrates ESG, CoT, and the stepwise confidence-driven memory bank. This configuration achieves the highest overall performance, with an average GCR of 41.68% and SR of 11.67%. The addition of the Memory component enables the model to retain and utilize high-confidence experiences, which significantly boosts performance in longer and more complex task sets, such as Median and Long.

**Memory Retention Evaluation:** To assess the impact of different evaluation methods for memory retention, we compared two approaches: evaluating each step of the CoT reasoning process versus evaluating the entire plan as a whole. As shown in Table 4, the stepwise evaluation consistently outperforms the whole plan evaluation across all task sets. When evaluating the entire plan (first row), the model achieves an average GCR of 37.29% and SR of 4.17%. While this method allows for a global assessment of task completion, it fails to capture finer-grained decision-making errors. In contrast, the stepwise evaluation method (second row) leads to a substantial improvement, with an average GCR of 41.68% and SR of 11.67%. By scoring each individual reasoning step, the model is able to identify and retain more high-confidence experiences. This granular approach helps the model refine its reasoning process at each stage, leading to better performance in overall task execution, as demonstrated by higher scores across all categories.

## 6 CONCLUSION

In this work, we introduced Memory-Driven Multimodal Chain of Thought (MCoT-Memory), a novel framework designed to address the challenges of long-horizon task planning in dynamic environments. By incorporating Evolving Scene Graph-Driven CoT and Stepwise Confidence-Driven Memory Retention, our approach enables agents to efficiently manage multi-step tasks by continuously updating visual representations and selectively retaining high-quality reasoning processes. These innovations allow MCoT-Memory to utilize long-term memory effectively and outperform existing methods. To comprehensively evaluate the performance of MCoT-Memory, we proposed ExtendaBench, a new benchmark specifically designed for long-horizon tasks. ExtendaBench consists of 1,198 tasks across four categories—ultra-short, short, median, and long—offering a diverse platform to rigorously assess task-planning models. Our experiments, conducted against several baselines, demonstrated that MCoT-Memory consistently enhances task success rates and goal condition recall, particularly in more complex, long-horizon scenarios. In summary, MCoT-Memory advances multimodal task planning and provides a strong foundation for future research in long-horizon task planning.

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

# A APPENDIX

## A.1 VISUALIZATION OF GENERATED DATA

### A.1.1 VIRTUALHOME

To better understand the structure and diversity of the tasks generated for VirtualHome, we provide visualizations of selected examples in Figure 3 and Figure 4. These figures illustrate the task environments and corresponding action sequences, demonstrating the complexity and variety of task settings created through GPT-4. The visualizations showcase the spatial arrangement of objects, agent interactions, and the multi-step nature of the tasks.

### A.1.2 HABITAT 2.0

We also provide visualization of selected example in Figure 5.

**Task: Prepare a nutritious breakfast by making a cereal bowl with slices of banana, accompanied by a glass of water and fork on kitchen table.**

Figure 3: Example of generated task in VirtualHome using GPT-4.

**Task: Prepare a festive fruit salad with a side of whipped cream and arrange a snack table with various items for a small gathering.**

Figure 4: Example of generated task in VirtualHome using GPT-4.

**Task: Please help me to transfer cup, book, bowl, strawberry, lego, banana from black table, black table and brown table to left counter and box , lemon from right drawer to sofa.**

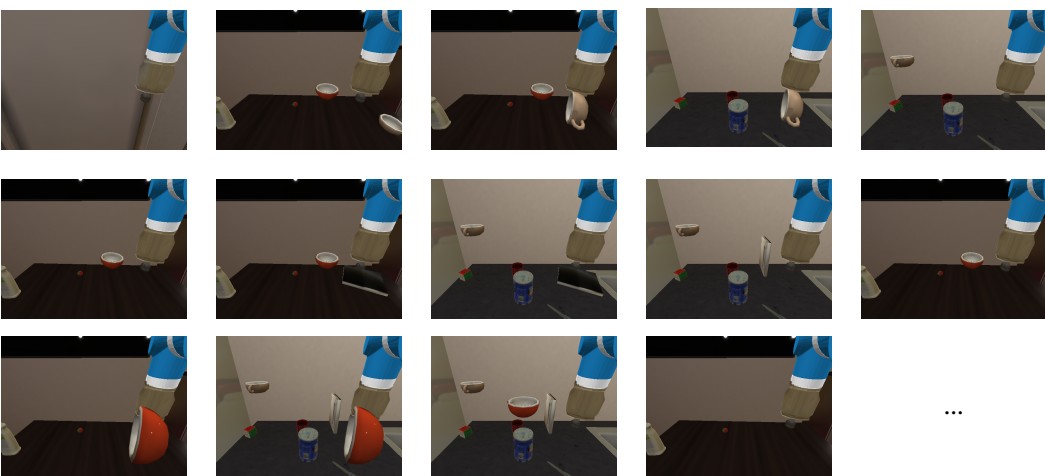

Figure 5: Example task in Habitat 2.0.

