# OpenReview forum: "Memory-Driven Multimodal Chain of Thought for Embodied Long-Horizon Task Planning"
_ICLR.cc/2025/Conference — ICLR 2025 Conference Withdrawn Submission_

### Official Review · Reviewer_MMgW · 2024-11-01

**Soundness:** 2
**Presentation:** 1
**Contribution:** 2
**Rating:** 3
**Confidence:** 5

**Summary:**

This paper introduces MCoT-Memory, a framework that integrates evolving scene graph-driven reasoning with stepwise confidence-driven memory retention, allowing robots to perform long-horizon, multi-step tasks in dynamic environments more effectively. Additionally, they propose ExtendaBench, a benchmark specifically designed to evaluate long-horizon tasks through a diverse set of 1,198 tasks, providing a robust platform for testing task-planning models. The authors implement several baseline models, and extensive experimental results demonstrate the significant performance improvements achieved by MCoT-Memory.

**Strengths:**

This paper demonstrates the potential of using Memory-Driven Multimodal Chain of Thought to solve Embodied Long-Horizon Task Planning and achieve good results.

This paper constructs a comprehensive evaluation system in VirtualHome and Habitat 2.0 to verify the effectiveness of the method.

**Weaknesses:**

Though it is a nice application to use LLM with a Memory-Driven Multimodal Chain of Thought for Embodied Long-Horizon Task Planning, I think the overall contribution of this paper is not enough to be considered for publication on ICLR:

1. Previously, many agents for long sequential tasks in embodied scenarios also used multiple memories, which were not discussed in this article, such as [1],[2],[3].

2. Similarly, there are many benchmarks for long-sequence tasks in embodied scenarios before, and the proposed benchmarks are not compared with these benchmarks in this paper. Further, I don't think it's convincing enough to just change the scene and set up a new benchmark。

3. The evaluation is limited. I think the author should compare it to the article in 1, not just to the baseline with no memory.

[1]Wang G, Xie Y, Jiang Y, et al. Voyager: An open-ended embodied agent with large language models[J]. arXiv preprint arXiv:2305.16291, 2023.
[2]Wang Z, Cai S, Chen G, et al. Describe, explain, plan and select: Interactive planning with large language models enables open-world multi-task agents[J]. arXiv preprint arXiv:2302.01560, 2023.
[3]Qin Y, Zhou E, Liu Q, et al. Mp5: A multi-modal open-ended embodied system in minecraft via active perception[C]//2024 IEEE/CVF Conference on Computer Vision and Pattern Recognition (CVPR). IEEE, 2024: 16307-16316.

**Questions:**

I would like to know more details about the execution phase, such as how the agent interacts with the simulation environment after generating the action to generate the next observation.

---

### Official Review · Reviewer_EUDw · 2024-11-03

**Soundness:** 2
**Presentation:** 2
**Contribution:** 2
**Rating:** 3
**Confidence:** 4

**Summary:**

The paper proposes Memory-Driven Multimodal Chain of Thought (MCoT-Memory), a framework to improve task planning performance of multimodal LLMs using: a.) a dynamic scene graph based representation of the environment that is updated online, b.) a confidence-driven memory retention mechanism that leverages a expert model to evaluate reasoning over time and only retaining experiences which are accurate and relevant. The proposed method combines scene graph based representation and a selective retention of experiences in the form of natural language descriptions to augment MLLMs with memory of past interaction and actions in an environment to improve task planning performance. Furthermore, to study long-horizon task-planning performance the paper proposes ExtendaBench a new benchmark with 1198 automatically generated tasks in 2 simulators by leveraging GPT4. Finally, the paper shows the proposed MCoT-Memory framework improves task planning performance of existing open source MLLM (llava) and GPT-4o on ExtendaBench.

**Strengths:**

1. The proposed approach of leveraging scene graph representation of the environment and interactions to augment MLLMs with history of interactions in the environment for embodied task is intuitive and a simple approach.
2. The idea of selectively retaining past experiences using Stepwise Confidence-Driven Memory Retention is also interesting and novel.
3. The proposed benchmark and automatic data generation pipeline is valuable to the community

**Weaknesses:**

1. The paper has a bunch of interesting ideas but they are not very well explained in the main paper and there is no additional supplementary available clarifying those details. For example, the dynamic updates to scene graph and construction in not very well explained and section 3.1.3 doesn’t have enough info. It’s unclear if all experiments presented used privilege information from the environment. Similarly, the memory retention logic is not very well explained. In section 3.2.1 authors mention there is a MLLM-based judge which is used to evaluate traces for retention but details about how memory traces are filtered out, what threshold was used, how was it decided, analysis around different thresholds is not provided.
2. The paper is also lacking in analysis of the proposed dataset. The authors discuss the data generation pipeline but there’s no analysis around the diversity of generated tasks, discussion around what are the type of tasks included in the dataset (are all tasks multiple object rearrangements or there are more complex tasks?), list of examples in supplementary is also missing (there are only 2 examples provided), or any human evaluations around dataset quality. It would’ve been great to provide more details about the dataset. From the examples provided in supplementary, the task used in habitat has some mistakes in language instructions which makes me concerned about dataset quality.
3. The results section is also a bit inconsistent. In table 2 only performance of GPT-4o is reported on habitat tasks and the proposed method only helps give marginal improvements on easiest version of tasks. This makes me concerned about how well the method works across baselines and environments. I would appreciate if authors can present results for LlaVA baseline and provide some analysis of failure modes for the proposed method.
4. The failure mode analysis for the proposed method is missing in the paper. It looks like all baselines perform quite poorly on the proposed benchmark so it would be good to have a section on analysis of failure modes that helps us understand why this task is hard and what are cases where current MLLMs fail even after augmenting with memory.

**Questions:**

In addition, I would also like to know more details about scene graph generation. Was there any privileged info used from the simulator while building the scene graph? In section 3.1.1. authors mention they use LLaVA to build scene graph but the scene graph has info about position, state, rooms, etc I would like to know how that information was extracted using LLaVA. My understanding is these models perform quite poorly for describing in simulated environments which could mean that quality of generated scene graph is quite poor which can explain poor performance on the task.

In the current state, I believe the paper needs quite a lot of improvement in writing and some more improves on experiments and analysis which is reflected in my rating. I would be happy to increasing my rating if authors address my concerns during rebuttal.

---

### Official Review · Reviewer_kH1q · 2024-11-04

**Soundness:** 2
**Presentation:** 3
**Contribution:** 2
**Rating:** 5
**Confidence:** 4

**Summary:**

This paper proposes a novel framework called Memory-Driven Multimodal Chain of Thought (MCoT-Memory) to enhance long-horizon task planning for embodied agents in dynamic environments. The key innovations are: 1) Evolving Scene Graph-Driven Chain of Thought with CoT Memory Retrieval, which allows the agent to continuously update a scene graph with visual information to inform real-time decision-making, and incorporate past experiences in its reasoning process; and 2) Stepwise Confidence-Driven Memory Retention, which evaluates the reasoning process and selectively retains high-confidence experiences to improve performance on long-horizon tasks. The authors also introduce a comprehensive benchmark, ExtendaBench, with 1,198 tasks across different horizon lengths to evaluate long-horizon task planning capabilities. Experiments demonstrate that MCoT-Memory significantly outperforms prior methods on long-horizon tasks.

**Strengths:**

1. This paper proposes a memory-driven CoT framework for embodied agents to plan long-horizen tasks. The problem and the method are elaborated clearly and the idea of maintaining a memory bank of  high-quality scene graphs and trajectories is reasonable.
2. A new task planning benchmark on VirtualHome and Habitat environments is developed. The task-generation process includes carefully reviewing and sequential refinement, which ensures the quality of the proposed benchmark.
3. Evaluation on the proposed ExtendaBench shows the effectiveness of the MCoT-Memory approach compared with baselines such as CoT, CCoT, and DDCoT. The ablation study also verifies the benefits of the method components including evolving scene graph, CoTm and memory bank.

**Weaknesses:**

1. Although the authors develop a new benchmark, there is a lack of introduction of other embodied task-planning benchmarks [1-4] and their comparison. What are the advantages and necessity of proposing ExtendaBench and their statistical evidence?
2. **The lack of experiments.** There should be experiments on other well-known task-planning benchmarks (even if they have deficiencies) and comparisons with other agentic frameworks tailored for task-planning [1] to better evaluate the proposed MCoT-Memory method. Also, the results of open-sourced VLMs on Habitat as well as an ablation study on Habitat are absent.
3. There should be an illustrated example of a scene graph defined in line 187. What are the tailored prompts for scene graph generation and updating introduced in section 3?
4. What are the criteria to determine the step-wise score and the overall score of the CoT trajectories? Are there any mechanisms to guarantee the objectiveness and fairness of such scores such as major voting, etc.?

[1] Wu Z, Wang Z, Xu X, et al. Embodied task planning with large language models[J]. arXiv preprint arXiv:2307.01848, 2023.
[2] Srivastava S, Li C, Lingelbach M, et al. Behavior: Benchmark for everyday household activities in virtual, interactive, and ecological environments[C]//Conference on robot learning. PMLR, 2022: 477-490.
[3] Gan C, Zhou S, Schwartz J, et al. The threedworld transport challenge: A visually guided task-and-motion planning benchmark for physically realistic embodied ai[J]. arXiv preprint arXiv:2103.14025, 2021.
[4] Zhang M, Hao J, Fu X, et al. MFE-ETP: A Comprehensive Evaluation Benchmark for Multi-modal Foundation Models on Embodied Task Planning[J]. arXiv preprint arXiv:2407.05047, 2024.

**Questions:**

Please response to the questions mentioned above.

---

### Official Review · Reviewer_wMNZ · 2024-11-05

**Soundness:** 3
**Presentation:** 2
**Contribution:** 2
**Rating:** 5
**Confidence:** 3

**Summary:**

This paper introduces the Memory-Driven Multimodal Chain of Thought (MCoT-Memory) framework to address long-horizon task planning challenges in dynamic environments. MCoT-Memory incorporates two key methods: (1) Evolving Scene Graph-Driven Chain of Thought, which updates the scene graph based on visual information, enabling contextual and real-time decision-making; and (2) Stepwise Confidence-Driven Memory Retention, which uses an expert model to evaluate and retain only high-confidence experiences. Additionally, the authors propose ExtendaBench, a benchmark designed to evaluate long-horizon planning with 1,198 tasks across different horizons. Experiments show MCoT-Memory’s improved performance on long-horizon tasks over prior methods.

**Strengths:**

1) The work addresses a meaningful gap in long-horizon task planning for embodied agents.
2) The experiments demonstrate advantage of MCoT-Memory over naively using LLMs.

**Weaknesses:**

1) Issues related to the benchmark:
  a) A more detailed comparison to prior benchmarks is required. What are the prior task planning benchmarks, what are the exact planning horizons used in prior work?
  b) What is the justification for choosing Virtual Home and Habitat 2 as the two benchmarks you built on top of? Is Habitat 2 providing different insights compared to Virtual Home?
2) Limited Novelty: MCoT-Memory combines existing techniques (scene graphs, Chain of Thought, memory retention) without any fundamentally new concepts. Neither does the paper provide any insights into why using scene graphs is helpful.
3) Missing experiments/ablations:
  a) On Habitat, only GPT-4o is evaluated. Any reason to not evaluate Llava/Qwen?
4) Typos/Mistakes:
  a) Starting of the abstract is abrupt: “Existing methods excel in short-horizon tasks”. What kinds of existing methods are they authors referring to?
  b) Opening quotation marks throughout the paper are facing backwards.
  c) Line 320: bu -> by
  d) Fig 2, Refinded Task and Plan box: The arrow is pointing in the opposite direction.
  e) Inconsistent use of symbols: ESG vs $\mathcal{E}_{\mathrm{SG}}$

**Questions:**

I have listed the questions in the bullet points of the weaknesses section.

---

### Note · Authors · 2024-11-14

I have read and agree with the venue's withdrawal policy on behalf of myself and my co-authors.